# Modulation of Anionic Lipid Bilayers by Specific Interplay of Protons and Calcium Ions

**DOI:** 10.3390/biom12121894

**Published:** 2022-12-17

**Authors:** Piotr Jurkiewicz, Martin Hof, Christoph Allolio, Jan Sýkora

**Affiliations:** 1Department of Biophysical Chemistry, J. Heyrovský Institute of Physical Chemistry of the CAS, Dolejškova 2155/3, 182 23 Prague, Czech Republic; 2Mathematical Institute of Charles University, Faculty of Mathematics and Physics, Charles University, Sokolovská 49/83, 186 75 Prague, Czech Republic

**Keywords:** phospholipid bilayer, calcium, proton, anionic lipids, headgroup organization, molecular dynamics, time dependent fluorescence shift, Laurdan, lipid hydration, headgroup mobility

## Abstract

Biomembranes, important building blocks of living organisms, are often exposed to large local fluctuations of pH and ionic strength. To capture changes in the membrane organization under such harsh conditions, we investigated the mobility and hydration of zwitterionic and anionic lipid bilayers upon elevated H_3_O^+^ and Ca^2+^ content by the time-dependent fluorescence shift (TDFS) technique. While the zwitterionic bilayers remain inert to lower pH and increased calcium concentrations, anionic membranes are responsive. Specifically, both bilayers enriched in phosphatidylserine (PS) and phosphatidylglycerol (PG) become dehydrated and rigidified at pH 4.0 compared to at pH 7.0. However, their reaction to the gradual Ca^2+^ increase in the acidic environment differs. While the PG bilayers exhibit strong rehydration and mild loosening of the carbonyl region, restoring membrane properties to those observed at pH 7.0, the PS bilayers remain dehydrated with minor bilayer stiffening. Molecular dynamics (MD) simulations support the strong binding of H_3_O^+^ to both PS and PG. Compared to PS, PG exhibits a weaker binding of Ca^2+^ also at a low pH.

## 1. Introduction

Lipid membranes are crucial building blocks of living organisms. They create effective barriers between the cell interior and the extracellular space, while at the same time dividing the cytosol into various compartments [1]. Biomembranes are exposed to very diverse conditions, for instance, local fluctuations of pH, temperature, and ionic strengths. Changes in acidity are of physiological importance, as the cell creates and maintains large pH gradients. Examples of very low pH (∼4–5) and high ion concentrations (∼mM) generated this way are the interior of vacuoles and lysosomes [2,3,4,5,6]. Furthermore, the periplasm of bacterial species adapted to extremely acidic environments also contains increased concentrations of protons (pH∼4–5) and other ions that interact with the bilayers bordering the periplasmic space [7]. As an example, the Gram-negative bacteria *Helicobacter pylori* survive in the highly acidic environment inside the stomach. Understanding all the aspects of its resistivity to low pH levels is of high interest as it causes serious gastric problems, threatening half of the world population [8].

The effect of pH on model lipid bilayers has been, therefore, studied extensively: in the case of negatively charged lipids, such as DOPS, pH was found to radically alter the curvature of the surrounding DOPE inverted hexagonal phase [9]. An example of a phase transition triggered by pH alone is found in cardiolipin [10]. In its ability to deliver dramatic changes to bilayers, the only comparable ion of physiological importance is Ca^2+^. Indeed, calcium and hydronium (i.e., hydrated proton H_3_O^+^) ions have been identified among the most potent ions in affecting the organization of phospholipid bilayers [11,12,13]. Interestingly, both cations show a similar modus operandi. They exhibit higher binding to negatively charged bilayers with a strong preference toward the headgroup region [14,15]. Specifically, the phosphate and carbonyls of the zwitterionic lecithins and carboxyl or phosphate groups of the negatively charged lipids serve as the binding partners [14,16]. The consequences of Ca^2+^ and H_3_O^+^ binding on bilayer organization are also similar, resulting in membrane stiffening, reducing the area per lipid, or altering the headgroup orientation [17,18]. The hydronium ion is in fact not a single species but forms a distribution, whose boundaries are defined by the Eigen and Zundel ions [17]. The content of this distribution can be shifted at interfaces. Furthermore, the hydronium ion can disappear by covalent binding to any negative charge and can reappear by dissociation, including the self-dissociation of water. The true distribution of H_3_O^+^ ions at any interface is not readily available, as is evidenced by the century-long discussion on the charge of the air–water interface [19]. Ca^2+^, on the other hand, is divalent, and thus can be expected to generate considerable charge densities. The Ca^2+^ ion is well-known for clustering lipids and inducing cross-links and fusion [20,21,22,23,24]. A particular quality of Ca^2+^ is its relatively weak binding to water oxygen atoms, which may thus be displaced by those of lipids. However, the interplay of Ca^2+^ and H_3_O^+^ is so far not well understood.

To obtain a more profound understanding of the membrane action under these atypical conditions, we investigated the pH-dependent Ca^2+^ effect on model membranes. Specifically, we focused on the lipid species commonly found in the lysosomal [4], vacuolar [5], and bacterial membranes [25]: 1-palmitoyl-2-oleoyl-sn-glycero-3-phosphocholine (POPC), 1-palmitoyl-2-oleoyl-sn-glycero-3-phosphoethanolamine (POPE), 1-palmitoyl-2-oleoyl-sn-glycero-3-phospho-L-serine (POPS), and 1-palmitoyl-2-oleoyl-sn-glycero-3-phospho-(1′-rac-glycerol) (POPG). The selected lipids differ in the headgroup moieties which largely determine the organization and arrangement of the self-assembled lipid bilayers. Zwitterionic PE lipid was reported to possess the lowest area per lipid of all the investigated phospholipids and a high propensity to create intramolecular hydrogen bonds between adjacent PEs [26,27]. Anionic lipids PS and PG show similar capabilities to bridge lipid molecules via hydrogen bonds [25,28]. Moreover, their negative net charge renders them an ideal target for cation binding [29,30]. In contrast, positively charged choline of zwitterionic PC bilayers interacts weakly with negatively charged phosphate or carbonyl groups of surrounding lipids, forming weak hydrogen bonds [31]. The simultaneous elevation of hydronium and calcium cation concentrations may potentially change the membrane fluidity and hydration in a headgroup selective manner.

In this study, we examine the interplay of Ca^2+^ and H_3_O^+^ adsorption on the headgroup organization on the POPE, POPS, and POPG bilayers. For this purpose, we monitor qualitative changes in a liposome surface charge with alternating pH and ionic strength by means of zeta potential measurements [32]. In order to map the changes in membrane organization at the carbonyl region, a time-dependent fluorescence shift (TDFS) analysis supplemented with generalized polarization (GP) is used [33]. By means of these experimental techniques combined with molecular dynamic (MD) simulations, a detailed understanding of ion–membrane interactions is obtained. In addition, we apply dynamic light scattering (DLS) [34] to map the extent of the vesicle aggregation.

## 2. Materials and Methods

*Materials:* 1-palmitoyl-2-oleoyl-sn-glycero-3-phosphocholine (POPC), 1-palmitoyl-2-oleoyl-sn-glycero-3-phosphoethanolamine (POPE), 1-palmitoyl-2-oleoyl-sn-glycero-3-phospho-L-serine (sodium salt) (POPS), and 1-palmitoyl-2-oleoyl-sn-glycero-3-phospho-(1′-rac-glycerol) (sodium salt) (POPG) were obtained from Avanti Polar Lipids, Inc. (Alabaster, AL, USA). 6-lauroyl-2-dimethylaminonaphthalene (Laurdan) and 1-[6-(dimethylamino)-2-naphthalenyl]-1-propanone (Prodan) were provided by Molecular Probes (Eugene, OR, USA). Calcium chloride (CaCl_2_), sodium hydroxide (NaOH), 2-[4-(2-hydroxyethyl)piperazin-1-yl]ethanesulfonic acid (HEPES), hydrochloric acid (HCl), acetic acid (CH_3_COOH), and sodium acetate (CH_3_COONa) were acquired from Sigma-Aldrich. 

*Liposome preparation: *Liposomes were prepared by adding appropriate volumes of lipids (POPC (100 mol%), POPC:PS (60:40 mol%), POPC:PG (60:40 mol%), POPC:PE (60:40 mol%), and Laurdan (1 mol%) in test tubes and drying them under nitrogen and then keeping them in vacuum for 2 h. Lipid films were rehydrated in de-ionized H_2_O (Milipore USA) containing either 10 mM HEPES buffer (pH 7) or 20 mM acetate buffer (pH 4). Large-unilamellar vesicles (LUVs) were then prepared via extrusion through 200 nm filters (Avestin, Ottawa, Canada). The final lipid concentration was kept at 1 mM with fluorescence probe concentration of 1:100 (mol/mol) with respect to the lipids. 

*DLS/Zeta potential measurements:* Dynamic light scattering technique was used to analyze the size of the liposomes prepared. Lipid samples were transferred to “dip” cell (UV grade poly (methyl methacrylate) cuvettes) (Malvern Instruments Ltd., Worcestershire, UK) to acquire zeta potential measurements. Samples of 0.2 mM lipid concentration were first equilibrated at 298 K for 3 min before each measurement. Zetasizer Nano ZS (Malvern Instruments Ltd.), consisting of a He–Ne laser (532nm) and an avalanche photodiode detector (APD), was used for light scattering experiments and the signal was collected at an angle of 173°. Size distributions and values of zeta potential were generated using Zetasizer 6.2 software (Malvern Instruments Ltd.). 

*Steady-state fluorescence spectroscopy:* Steady-state spectra were acquired using an Edinburgh FS5 Spectrofluorometer with Xenon-arc lamp. Samples were transferred to 1.4 mL quartz cuvettes (Hellma), which were incubated for 5 min at 298 K. Temperature was maintained via water circulation bath. Excitation and emission spectra were recorded with the emission wavelengths of 440 and 490 nm, and excitation wavelength of 373 nm, respectively. Generalized polarization was obtained according to Parasassi et al. [35]:(1)$$GP=\frac{I_{em}^{440}-I_{em}^{490}}{I_{em}^{440}+I_{em}^{490}}$$

*Time-dependent fluorescence shift (TDFS) method:* Time-correlated single-photon counting (TCSPC) data were acquired using 375 nm laser (LDH-D-C-375, Picoquant, Germany) and IBH 5000U fluorescence spectrometer with cooled Hamamatsu R3809U-50 microchannel plate photomultiplier (Hamamatsu, Japan). Decays were collected in the range of the emission wavelength from 400 nm to 540 nm with an increment of 10 nm. An emission cut-off filter (cutoff: > 399 nm) was inserted into the detection arm of the set-up. An iterative re-convolution procedure was used to fit the measured multi-exponential decays using IBH-DAS6 software. The spectral reconstruction combining steady-state and TCSPC data generated time-resolved emission spectra (TRES) which were fitted by a log-normal function [36]. The gained time-course of the TRES maxima *ν*(*t*) was used to calculate relaxation time *τ_r_* and overall dynamic Stokes shift Δ*ν* [37]:(2)Δν=ν(0)−ν(∞)
(3)$$\tau_{r}=\int_{0}^{\infty }\frac{\nu \left(t\right)-\nu \left(\infty \right)}{\Delta \nu }dt$$
where the position of the maximum of the spectrum emitted at „time 0“ *ν*(0) was estimated by means of the procedure described in [38], and *ν*(∞) was obtained by extrapolation of the *ν(t)* dependence for *t* → ∞. 

*Molecular dynamics:* MD simulations were performed using the CHARMM36 [39] force field for membranes and the GROMACS 2020 software [40]. We used an NpT ensemble at 1 bar and 303.15 K, using semi-isotropic pressure coupling, via a Parrinello-Rahman barostat [41]. Temperature was controlled using a Nosé–Hoover thermostat [42]. The timestep was 2 fs, we used particle mesh Ewald (PME) electrostatics [43] with a cutoff of 1.2 nm and force switching for LJ interactions between 1.0 and 1.2 nm. Systems were prepared using CHARMM-GUI [44]. Then, the water model was switched to SPC/E [45], followed by a pre-equilibration for 200 ns. The production run was also 200 ns long. For the charged lipid mixtures, we used 130 lipids (65 total and 26 anionic lipids per leaflet, corresponding to the 40 mol% anionic lipid concentration), ca. 11K waters, and H_3_O^+^ ions from a Kirkwood–Buff force field, optimized for SPC/E [46] and proven to work at the air–water interface [47]. Modeling the protonation via the presence of H_3_O^+^ assumes preference for the Eigen cation and the absence of covalently bound protons. The systems we simulated contained a 0.15 M KCl concentration. In addition, we added neutralizing K^+^ counterions, to compensate lipid charge. We (partially) exchanged these counterions either with H_3_O^+^, Ca^2+^, or an equal mixture of the two, where equality was set by concentration, i.e., 26 H_3_O^+^ ions, 26 Ca^2+^ or, for the mixed system 17 Ca^2+^ and 18 H_3_O^+^. In total, six anionic lipid systems (containing POPC, POPG, and POPS) and a POPC system were prepared. Our simulations included the NBFIX [6] corrections as currently available in CHARMM-GUI.

## 3. Results and Discussion

**Dynamic light scattering (DLS) and zeta potential measurements:** DLS data are summarized in Appendix A. Number weighted size distributions indicate that liposome aggregation did not become dominant at any of the investigated Ca^2+^ concentrations. Nevertheless, a larger scatter in the curves detected for DOPC:PS suggests the higher polydispersity of these liposomes compared to the other lipid compositions. Zeta potential profiles of different lipid compositions and their dependence on pH and Ca^2+^ concentrations are shown in Figure 1. At the absence of calcium and at a neutral pH, the liposomes containing anionic lipids PS and PG possess lower zeta potential compared to the zwitterionic POPC and POPC:PE compositions. This finding confirms the overall negative surface charge of the vesicles carrying anionic lipids. Moreover, the zeta potential of PS containing vesicles is shifted to lower values than those containing PG. Upon pH decrease to 4.0, the POPC, POPC:PE, and POPC:PG liposomes exhibit only a minor increase in the zeta potential while the POPC:PS system exhibits a significant elevation shift in its zeta potential value by approximately +30 mV in comparison to the neutral pH. The protonation of the carboxylate of PS headgroup (pKa ∼ 4.4–5.5 [48,49]) is likely to be responsible for this substantial increase [32]. The addition of Ca^2+^ cations is accompanied by similar effects for all the investigated lipid systems. The gradual increase in the zeta potential takes the place of up to 200 mM concentrations of Ca^2+^. At this threshold, due to overcharging [50], all the lipid systems possess comparable potential regardless of the lipid composition and pH. This phenomenon, typical for high calcium loads, has been reported and discussed in more detail elsewhere [20]. The observed trends agree well with those previously reported in the literature: surface charge and zeta potential gained for the liposomes carrying negatively charged lipids are shifted to lower values compared to the bilayers composed of zwitterionic lipids [49]. Within the negatively charged bilayers, the PS headgroup decreases the zeta potential more profoundly than lipids with PG headgroups [51]. In addition, the presence of divalent cations generally leads to the gradual increase in the zeta potential reaching zero- or slightly positive values for millimolar Ca^2+^ concentrations [52]. In the context of the below presented TDFS data, the dipole potential may be of higher relevance compared to the recorded zeta potentials [53,54]. However, its determination by means of the conductivity measurements of hydrophobic ions is laborious and its interpretation at the presence of Ca^2+^ cations is rather challenging [55].

**Generalized polarization (GP) and time-dependent fluorescence shift (TDFS) in POPC, POPC:PE, POPC:PG, and POPC:PS bilayers**. In general, an important prerequisite for the quantitative application of fluorescence techniques in biomembrane research is the knowledge of the depth of the fluorescence probe (i.e., Laurdan) inside the bilayer. Inevitably, the decrease in pH can result in the protonation of the dimethylamino moiety of Laurdan, which could potentially alter its positioning along the bilayer normal, causing the data misinterpretation. To rule out this potential artifact, we monitored the effect of pH on the emission characteristics of Prodan—a water-soluble analogue of Laurdan containing an identical fluorophore. As is evident from Appendix A, the gradual decrease in pH causes the loss of the fluorescence emission. This substantial reduction in the fluorescence signal can be attributed to the protonation of the dimethylamino group, which strongly promotes the non-radiative pathways of excited state deactivation as anticipated by molecular modeling [56]. Due to the low fluorescence of the protonated dimethylamino–naphtalene fluorophore, we believe that Laurdan protonation influences the recorded fluorescence signal only marginally and does not cause artifacts in the data interpretation. 

The environment-sensitive behavior of Laurdan was demonstrated as a useful indicator of the bilayer organization. In the simplest concept, a steady-state “ratiometric” parameter called generalized polarization (*GP*) [57] is utilized, providing a measure of the changes in lipid mobility and lipid hydration at the sn-1 carbonyl region. The *GP* data obtained for the investigated lipid compositions are shown in Figure 2. Apparently, zwitterionic lipid systems remain intact at elevated Ca^2+^ and H_3_O^+^ concentrations (Figure 2A). POPE-containing liposomes possess higher *GP* values compared to pure POPC vesicles. This can be rationalized by the higher packing of PE headgroups [23]. The *GP* obtained for vesicles carrying the negatively charged lipids POPC:PS and POPC:PG appears to be more responsive to both pH and Ca^2+^ content. First of all, a drop in pH decreases the *GP *substantially, leading to decreased mobility and/or dehydration within the headgroup region (Figure 2B). Moreover, POPC:PG reacts differently to the elevated Ca^2+^ content compared to POPC:PS. *GP, *for the latter composition, remains constant in the acidic environment and increases at a neutral pH, which also holds true for POPC:PG. However, at a low pH of 4.0, liposomes composed of POPC:PG follow an unexpected *GP* evolution and the presence of Ca^2+^ leads to a gradual drop in *GP*. In order to elucidate this phenomenon, TDFS is a more appropriate method of choice than *GP,* as clarified below. 

*GP* measurements were also used for the assessment of the non-specific effects caused by the interaction of the counterions with the lipid headgroups [17]. In order to confirm that the observed trends are dominated by the H_3_O^+^ and Ca^2+^ cations, we monitored the GP dependence in the buffered systems (i.e., acetate and HEPES buffers for pH levels 4.0 and 7.0, respectively) as well as the solutions containing only Cl^-^ counterions (Figure 2C). The recorded data show minor mutual shifts of the measured curves; nevertheless, the main trends and features are common for both buffered samples and samples containing merely Cl^-^ anions. Therefore, the nature of the counterions appears to modulate the recorded fluorescence data to a minor extent. 

Another prominent feature of Laurdan is its sensitivity to the phase state of the lipid bilayer. Namely, the line-shape of its excitation spectrum reflects the actual phase of the membrane, being also responsive to the phase co-existence [58]. As shown in Appendix A, the line shapes of Laurdan embedded in all the investigated lipid compositions upon the elevated levels of H_3_O^+^ and Ca^2+^ remain virtually unchanged in respect to the neutral pH and calcium-free samples. Therefore, we conclude that the phase state of the lipid membranes was not altered under any of the applied conditions.

In spite of the broad use of the *GP* parameters described above [59,60,61], its interpretation can easily become precarious, since it reflects both the changes in the membrane mobility and hydration simultaneously in an indistinguishable manner [62]. Therefore, a more rigorous approach is needed in the application of a time-resolved technique that can separate the particular contributions of those factors. Namely, time-dependent fluorescence shift (TDFS) is a method of choice, providing the relaxation time (*τ_r_*) and overall dynamic Stokes shift (*Δν*) as a measure of mobility and extent of hydration within the carbonyl region, respectively. 

With the above-described effort to minimize the undesired effects of Laurdan protonization and counterion interference, the TDFS data were recorded for various lipid compositions at different pH levels and Ca^2+^ contents (Figure 3). Vesicles bearing the zwitterionic lipids POPC and POPC:PE appear to be insensitive to the explored factors and both parameters *τ_r_* and Δ*ν* do not exhibit significant differences within the experimental error at both the investigated pH levels at the explored Ca^2+^ range (Figure 3B,D). The presence of Ca^2+^ leads only to a slight decrease in the mobility of the Laurdan microenvironment, which is an effect already observed at pH 7.5 [20]. A completely different picture is observed for the liposomes encompassing anionic lipids (Figure 3A,C). At a neutral pH, the relaxation time becomes retarded even for the lowest concentration of Ca^2+^, which proves the importance of the electrostatic interactions for the ion–membrane interactions (Figure 3A). The further addition of Ca^2+^ ions leads to the additional increase in the membrane rigidity, yet it is not as noticeable as at 10 mM Ca^2+^. The Ca^2+^ dependencies of both *τ_r_* and Δ*ν* for POPC:PS and POPC:PG appear to run in parallel, resembling the mode of action of the Ca^2+^ for both the anionic vesicles, which again highlights the electrostatics as the driving force for the mutual ion–membrane interaction.

The reduction in pH to 4.0 triggers substantial changes in the headgroup organization both for POPC:PG and POPC:PS. Please note that the latter liposomes undergo protonation of the serine carboxylate, causing partial loss of their anionic character as proven by the zeta potential measurements. In spite of this fact, the elevated proton concentration modifies the headgroups of POPC:PG and POPC:PS in a similar manner, dramatically reducing the level of hydration as well as mobility within the carbonyl region. At the elevated Ca^2+^ concentrations, however, POPC:PG follows a different behavior compared to POPC:PS. Liposomes containing phosphatidyl–glycerol seem to be remarkably responsive to the presence of Ca^2+^, which leads to the decrease in *τ_r_* and elevation of Δ*ν*. This finding demonstrates that the increase in the headgroup fluidity and hydration upon the Ca^2+^ addition is unusual, and in fact a completely opposite effect is to be expected due to the potency of Ca^2+^ to bridge and cluster the anionic lipids. This anomalous behavior is observed only in the lower range of calcium concentrations (2.5 mM–100 mM) and diminishes at 200 mM Ca^2+^ content. Factually, for these limiting Ca^2+^ concentrations, the proton impact on the bilayer organization becomes minimized as the Δ*ν* and *τ_r_* dependencies obtained for the investigated pH levels come closer to each other. 

The gradual addition of Ca^2+^ cations to POPC:PS liposomes at pH 4.0 shows different modes of action compared to POPC:PG. Particularly, the protonation of the PS headgroup at pH 4.0 appears to shield the effect of Ca^2+^ cations observed for the POPC:PG, since the relaxation time *τ_r_* increases with elevated Ca^2+^. In parallel, the overall Stokes shift Δ*ν* shows only a slight up-rise at the medium calcium concentrations followed by the subtle drop at the highest Ca^2+^ content. However, the observed changes are close to the TDFS detection limit and cannot compete with those observed for POPC:PG liposomes. 

**MD simulations**: Simulations were conducted using H_3_O^+^ ions in SPC/E water. This means that Zundel ions and the covalent binding of protons to lipids are not part of the model. The use of the SPC/E water model is motivated by the parametrization of the H_3_O^+^ forcefield for this potential as well as the bad performance of the TIP3P model in the description of membrane surface tension [63]. We simulated a POPC bilayer in SPC/E water, obtaining an area per lipid (APL) of 0.643 nm^2^, which is at the lower limit of the experimental values [64]. In general, it seems that the SPC/E model leads to somewhat thicker membranes than the TIP3P model when using the CHARMM force field. Our main focus is the comparison of the effects of Ca^2+^ and H_3_O^+^, so that we accept these limitations of the simulation accuracy. Furthermore, we do not know the chemical potential of either the ions or water, so we cannot make a direct comparison to the experimental conditions. This limitation is common in MD simulations, but it nevertheless should be pointed out from time to time. In order to rigorously compare to an experimental concentration, it would be necessary to run a grand-canonical simulation, allowing for the exchange of ions and water with an external bath of defined concentration. Its chemical potential would then have to correspond to the experimental calcium concentration, as the simulation lipid concentration (appr. 0.4 M) is high compared to the experimental one. 

Comparing the effects of Ca^2+^ and H_3_O^+^ on the anionic membranes, we find that both ions bind strongly to the anionic lipids. In Figure 4A,B, the symmetrized density profiles at the lipid–water interface are shown for the mixed simulations, where both Ca^2+^ and H_3_O^+^ are present. In addition, we show MD snapshots. It is evident that both ions bind more strongly to the headgroups than K^+^ and Cl^-^ ions, as is expected. We find H_3_O^+^ to bind at a deeper level than the Ca^2+^ ions. Both ions are close to the position of the phosphate group, but Ca^2+^ appears to bind closer to the interface. It is clear that the binding patterns of both ions will be complex. In our MD snapshot shown in Figure 4, we find Ca^2+^ to be particularly well bound to the PS carboxyls, whereas the H_3_O^+^ ion is mainly coordinated via hydrogen bonds to the phosphate group. For PG, both ions seem to coordinate mainly to the phosphates. The strong preferential binding of Ca^2+^ to the carboxyls, in the case of PS, may be due to the used NBFIX correction, which is state-of-the-art for CHARMM. Recently improved force fields using ECC corrections have been proposed by the Jungwirth group [65]. A compatible H_3_O^+^ potential has not been released yet.

When comparing POPC:PS to POPC:PG, we find that both Ca^2+^ and H_3_O^+^ are more strongly bound to POPC:PS than to POPC:PG. The evidence is shown in the peak heights in Figure 4A,B. On POPC:PG, Ca^2+^ binding is weak enough for regular dissociation on the simulation timescale. It is worth noting that, in a comparable simulation of pure POPC, with a different water model the binding of Ca^2+^ to PC was very weak [21]. Furthermore, it appears that in POPC:PS the H_3_O^+^ peak is enhanced by the presence of Ca^2+^ (see Figure 4C) and the protons are not displaced. An equivalent observation does not exist in the POPC:PG mixture, where there is no such effect, only a small weakening effect on Ca^2+^ binding conditional to proton adsorption is possible. The trends in the area per lipid (APL) are slightly different. They are shown in Figure 4D. As for the PS system, the area per lipid is consistently low in the presence of both ions, and the PG system displays a stronger effect of the equivalent number of H_3_O^+^. 

Please note that salt and lipid concentrations in MD simulations are usually very different from those of model experiments, e.g., the bulk concentrations in our simulation of PS are very low (compare the right half of the density profiles in Figure 4), as we do not observe dissociation of ions, while in the case of PG the calcium concentration in the bulk is comparable to the lower end of those used in the experimental section. 

In general, physiological conditions cannot be fully reproduced in model studies. Calcium-signaling requires that the concentration of calcium cations be strictly regulated in the cell. In the cytosol, it varies between nanomolar and micromolar (1–2 μM), but in the endoplasmic reticulum it can already exceed 500 μM and locally the calcium concentration can be increased to millimolar (calcium levels in the extracellular fluid are ~2 mM) [66]. In this work, we use higher calcium concentrations both in our experiments (2.5–200 mM) and in the MD simulations, to be able to observe the global calcium effects on the properties of model lipid systems. We are limited by the sensitivity of the chosen methods but also by our simplified model systems. Therefore, the obtained results cannot be directly linked to the living systems in a simple one-to-one fashion. Instead, we describe the mechanisms of Ca^2+^ and H_3_O^+^ ion adsorption to synthetic phospholipids that mimic the natural membranes that are exposed to the acidic environment with higher ion concentrations. Zwitterionic PC and PE are the dominant phospholipids in vacuoles of both plant and yeast [67]. For the negatively charged membranes, we used the lipids-comprising PG headgroup frequently found in lysosomal bilayers [68] and periplasmic membrane Gram-negative bacteria [69], and PS as a control. Lysosomal membranes contain sphingomyelin, cholesterol, and the uncommon negatively charged lipid, bis(monoacylglycero)phosphate (BMP) [68], which is absent in other cellular membranes. Herein, however, we had to limit our study to the most common phospholipids. The mechanisms that we describe may be important for the interpretation of the biological processes, but their applicability to certain biological conditions should be evaluated. 

## 4. Conclusions

We mapped the mobility and hydration within the carbonyl region of lipid bilayers with different headgroup compositions at elevated calcium and hydronium cation concentrations. TDFS data showed that the carbonyl region of the zwitterionic bilayers composed of POPC or POPC:PE was insensitive to either H_3_O^+^ or to Ca^2+^ presence. In contrast, the anionic bilayers with the compositions POPC:PS and POPC:PG appeared to be highly responsive to both H_3_O^+^ and Ca^2+^, each of them in a different manner. At a low pH and in the absence of Ca^2+^, both bilayers become dehydrated and rigidified. Upon the Ca^2+^ addition at pH 4.0, the POPC:PG bilayers restore their hydration and fluidity, almost reaching the level observed for neutral pH. In contrast, in POPC:PS bilayers, the Ca^2+^ presence at acidic pH is barely detectable and only a mild stiffening of the membrane is noticed. We speculate that the protonation of the PS serine provoked by lower pH shields the effects, which are commonly observed upon Ca^2+^–membrane interaction [14]. MD simulations predict the reduction in the area per lipid for both anionic lipid systems in acidic environments, which agrees with the experimental finding of a significant decrease in bilayer mobility at pH 4.0. Moreover, the strong affinity of the investigated ions to POPC:PS bilayers found in the simulations corresponds well to the TDFS profiles of POPC:PS liposomes at low pH. Under these conditions, added Ca^2+^ cannot substitute the strongly interacting H_3_O^+^ ions. In contrast, simulations suggest that the binding of Ca^2+^ to PG is weaker than PS. In the experiments, at pH 4.0, both the bilayer hydration and mobility of DOPC:PS are affected by the gradual increase in Ca^2+^, to a lesser extent compared to the DOPC:PG membrane. We believe that these findings can bring a deeper understanding of the biophysical behavior of the lysosomal, vacuolar, and bacterial membranes and shed a light on the fate of drug nanocarriers under diverse conditions occurring in living organisms [70].

## Figures and Tables

**Figure 1 biomolecules-12-01894-f001:**
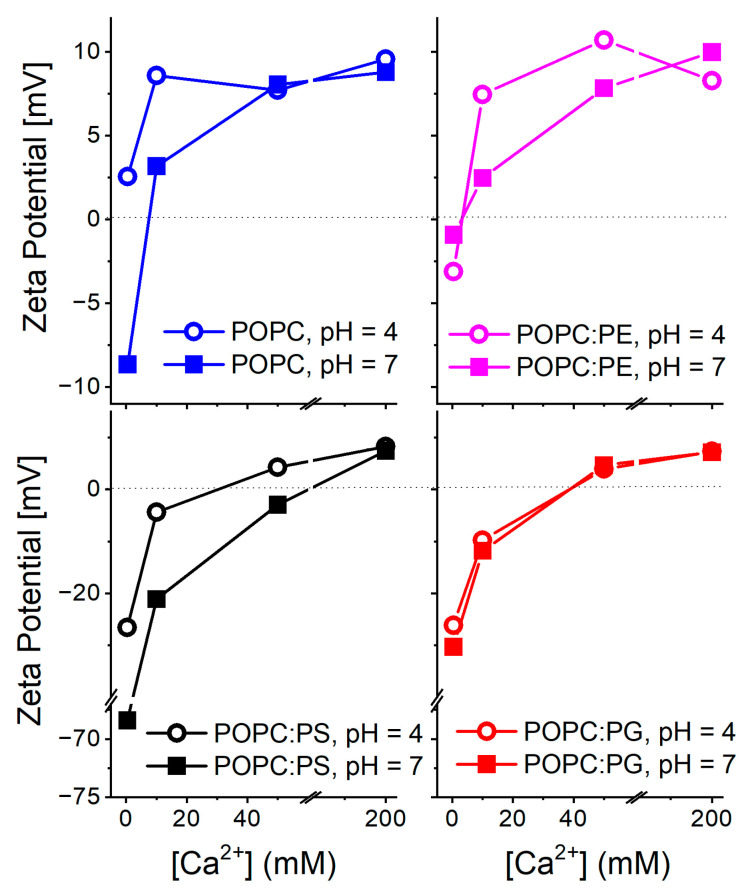
Profiles of zeta potential obtained for different lipid compositions at various pH levels and Ca^2+^ concentrations. The left and right panels show data for the liposomes containing 40% of the anionic lipids POPS and POPG. The left panel summarizes the data gained for the zwitterionic lipids POPC and POPC with 40% of POPE. Please note the different scaling of the zeta potential axis of the bottom panels. Values obtained for 200 mM Ca^2+^ possess the largest error of all the recorded values due to the intrinsic uncertainty of the Zetasizer measurements at high ion concentrations [32].

**Figure 2 biomolecules-12-01894-f002:**
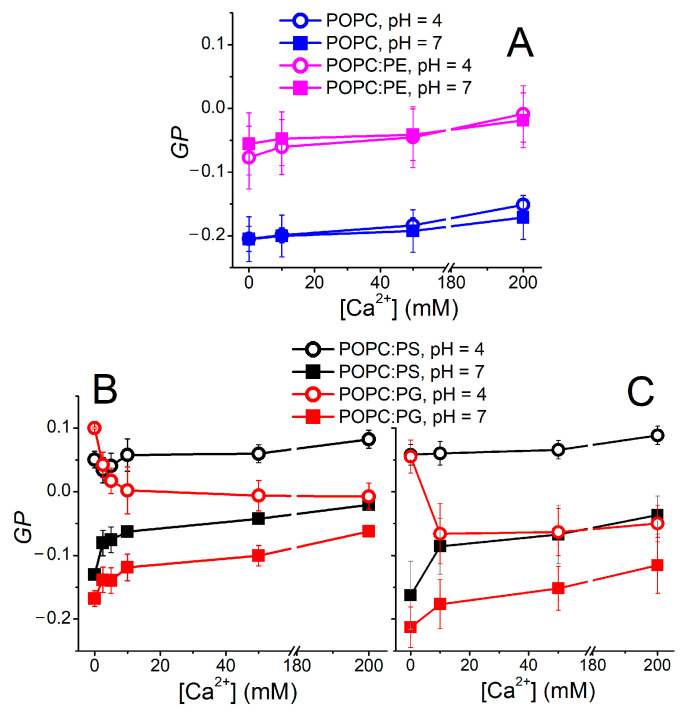
Ca^2+^-dependent changes in GP parameters measured at pH 4.0 and 7.0. Top panel (**A**) shows *GP* evolution recorded for zwitterionic lipids POPC and POPC:PE. Bottom left panel (**B**) depicts the data recorded in 10 mM HEPES (pH 7.0) and 20 mM acetate buffers (pH 4.0). Bottom right panel (**C**) illustrates the effect of counterion on *GP*. It shows the same dependence as in panel B; nevertheless, the desired pH values were reached by the addition of NaOH or HCl (pH indicator fluorescein was used to determine the required aliquots of NaOH/HCl to reach the desired pH). Similar course of the data in panels B and C documents the minor role of the counterions. Please note that GP is not a suitable parameter for quantitative characterization of the Ca^2+^ and H_3_O^+^ effects on the bilayer hydration and fluidity, since both these factors change simultaneously (hydration (Δν) and mobility (*τ_r_*) parameters in Figure 3A,C). Yet, GP can serve as a useful qualitative measure of the counterion impact on the membrane organization.

**Figure 3 biomolecules-12-01894-f003:**
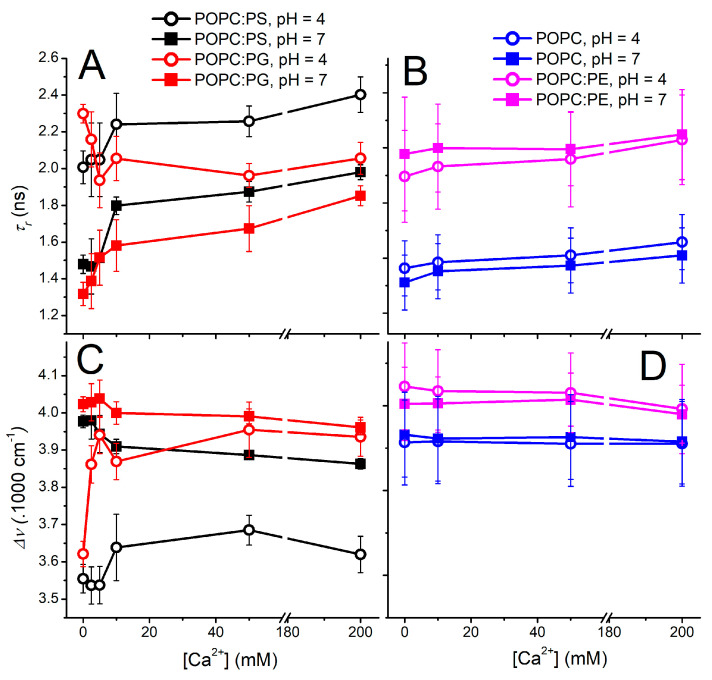
Dependence of the relaxation time *τ_r_* (panels (**A**,**B**)) and overall Stokes shift Δ*ν* (lower panels (**C**,**D**)) on the calcium concentration obtained for various lipid compositions at pH 4.0 (open symbols) and 7.0 (solid symbols). *τ_r_* and Δ*ν* profiles of the POPC liposomes containing 40% of anionic lipids POPS and POPG are summarized in left panels (**A**,**C**), while those for zwitterionic systems (POPC and POPC with 40% of POPE) are shown in panels (**B**,**D**). All samples were measured at 25 °C.

**Figure 4 biomolecules-12-01894-f004:**
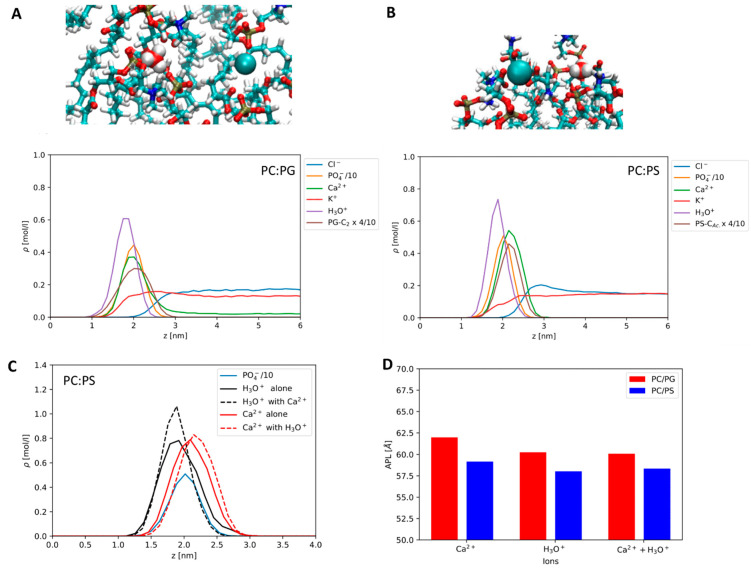
Molecular dynamics results: (**A**) Top: MD snapshot showing binding of Ca^2+^ and H_3_O^+^ to headgroups of PC and PG, mainly PO_4_^−^. Bottom: Density profile for the PC:PG membrane in the H_3_O^+^/Ca^2+^ mixture, including scaled PO_4_^−^ and PG glycerol C_2_ densities. The zero on the z axis refers to the membrane center. (**B**) Corresponding snapshot and profile for the PC:PS mixture. Here, we use the scaled PS carboxyl C atom for the headgroup density. (**C**) Density profile of Ca^2+^ at the PC:PS interface in presence and in absence of H_3_O^+^, Ca^2+^ with ion densities for the mixture scaled to the relative number of ions in the single-cation simulations to facilitate comparison. (**D**) Areas per lipid (Å^2^) for the membrane mixtures in presence of H_3_O^+^, Ca^2+^, and their mixture.

## Data Availability

Data can be shared upon request from the corresponding author.

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
