# Peer review of "Modulation of Anionic Lipid Bilayers by Specific Interplay of Protons and Calcium Ions"

_biomolecules, 2022, doi:10.3390/biom12121894_

Round 1

Reviewer 2 Report

Abhinav et al report a nice experimental and computational work on the effects of Ca2+ and H3O+ on liposomes composed of pure POPC, or mixtures 60:40 mol % of POPC with POPS, POPG or POPE. This is a very relevant topic for understanding the behavior of biological membranes under physiological and pathological conditions. Experiments and simulations included in the present report are well designed and performed, and conclusions are supported by the presented data. However, there are some important points that need to be addressed in a revised version before this manuscript is suitable for publication in Biomolecules.

Mayor points

1) The section on the effect of Ca2+ and H3O+, evaluated by fluorescence techniques, is somewhat confusing and, in my opinion, it needs to be rearranged in the revised version.

Results measuring the generalized polarization of Laurdan are partially included in the supporting information, and are not analyzed in terms of Ca2+ and H3O+ in the main text. I suggest that first panel in Figure S3 be moved to the body of the article as a new Figure 2, including new panels containing new data on PC and PC:PE liposomes (I assume that the authors have not included these panels because no significant differences were found. Anyway, these results have to be shown). After this new figure, the authors have to discuss it in terms of effects of Ca2+ and H3O+ on Laurdan GP, and then discuss the problems with the interpretation of GP results.

This could then connect with the experiments measuring the time-dependent fluorescence shift in Laurdan fluorescence, showing how this new technique overcomes the aforementioned problems in interpreting generalized polarization data. The analysis must first include a description of which changes are statistically significant. For the end, the discussion of results in terms of hydration and mobility is OK.

2) The section on molecular dynamics have to include new figures in the revised version. In page 8 lines 258-259 the authors said “Comparing the effects of Ca2+ and H3O+ on the anionic membranes, we find that both ions bind strongly to the anionic lipids.” In my opinion this sentence requires to shown a new figure, similar to panels A and B of Figure 2, including the information about POPC and POPC:POPE (this new figure could be included in the supporting information.

On the other hand, the discussion about the molecular events related to the effects of Ca2+ and H3O+ on anionic membranes would be clearer if a new figure is included showing two snapshots of the simulations detailing the coordination of Ca2+ and H3O+ with the headgroups of phosphatidylserine and phosphatidyl glycerol.

Minor points

3) Please check if the first or last name of the first author is missing.

4) Page 3 lines 90-97. The fluorescent probe PRODAN was not included.

5) Page 3 lines 99 and 100. Consider to replace the sentence “Liposomes were prepared by adding appropriate volumes of lipids (POPC (100%), POPC:PS (60:40 mol %), POPC:PG (60:40 mol %), POPC:PE (60:40 mol %) and fluorescent dye …” by “Liposomes were prepared by adding appropriate volumes of lipids (POPC (100%), POPC:PS (60:40 mol %), POPC:PG (60:40 mol %), POPC:PE (60:40 mol %) and Laurdan …” and indicates the approximated proportion of Laurdan in the liposomes (in mol %)

6) Page 3 line 120. The abbreviation TDFS is not spelling out

7) Figure legends are very short and lack relevant information. Please include all the information for understanding the full details in each panel of the figures.

8) Figure 1 can be more easily understood if it is split into four panels, one for each phospholipidic mixture

9) Page 6 line 206. The authors said “zwitterionic lipids POPC and POPC:PE are sensitive to the pH to a lesser extend compared to …”. Please revise this sentence. Given the error bars shown in Figure 2, it appears that these lipids are not significantly sensitive to either H3O+ or Ca2+.

10) Page 8 Figure 3. The reference point selected for the zero value of the x-axis in panels A, B and C is not clear. Besides, is the y-axis unit in these panels mole per liter? If this were true, these values have to be discussed in comparison to the very low physiological concentrations of H3O+ and Ca2+.

11) Page 2 of the Supplementary information. The sigmoidal fit in pH curves is not a good option for estimating pKa values. The sigmoidal shape is an artifact of using a logarithmic scale in the x-axis. To evaluate the dissociation constant it is better to fit a hyperbolic function (in the case of a single binding site or identical and non-interacting sites) to the data of integrated fluorescence intensity as a function of the free H3O+ concentration.

Reviewer 3 Report

I read the manuscript by Jan Sykora and coworker with great interest. The effect of calcium or pH on vesicle fusion has been studied since last century (see D. Deamer, A. Bangham and countless others). Zetapotential and light scattering measurements has been published many times. In the recent years MD simulation started to provide a molecular view on whats going on at the interface. A combination of both is indeed timely, a combination of experiment and modeling might be very useful for the community.

However, reading the manuscript I found that this is too premature and not yet at the level to provide a significant improvement.

The author should substantially more careful revising the literature. I know this is very difficult but not an excuse to ignore them.

Usually the Zetasizer doesn t allow to measure at small and higher salt concentration, how did you protect the electrode at such high salts?

There are standard fluorescence experiments to distinguish fusion from adsoption

The figure 3: the zero is the middle o the membrane? Maybe one could add the structure of the headgroup?

Fig S1: How reproducible is the data (I see the error bars; how did you get those?)

Please give more background information on the MD modelling

Comparision with older experimental data: One series of experiments have been done on the surface potential (see H.L. Brockman, for example PNAS 85 (1988) 4285 but many more by him and others)

In conclusion: this is a nice starting point of revisiting old observation but too premature at this point.

Round 2

Reviewer 2 Report

The authors have addressed all the issues raised by this reviewer in the first round of comments. The revised version of the manuscript has been substantially improved. Thus, I recommend its publication in Biomolecules.

Reviewer 3 Report

I think the authors improved the manuscript and it can be published as it is.

One point about Brockman, sorry that I insist but some years ago (almost 30!) this has been a topic: the orientation of lipid dipoles gave raise to a large surface potential. Brockman did monolayer measurements but also recorded the surface potential at the air-water interphase. He published a few very systematic papers. I think that there is quiet some information in as it is a different geometry. The conclusion which I took from those experiments was that there is no quantitative agreement between the model systems. Strange? Please have a look.   

 Chem Phys Lipids 1994 vol 73 p 57-79

doi: 10.1016/0009-3084(94)90174-0. (and a few more)
